# Looking for Targets to Restore the Contractile Function in Congenital Myopathy Caused by Gln^147^Pro Tropomyosin

**DOI:** 10.3390/ijms21207590

**Published:** 2020-10-14

**Authors:** Olga E. Karpicheva, Armen O. Simonyan, Nikita A. Rysev, Charles S. Redwood, Yurii S. Borovikov

**Affiliations:** 1Institute of Cytology, Russian Academy of Sciences, 4 Tikhoretsky Av., 194064 St. Petersburg, Russia; simonyan_armen@mail.ru (A.O.S.); nikrysev@gmail.com (N.A.R.); borovikov@incras.ru (Y.S.B.); 2Department of Biophysics, Faculty of Biology, Saint Petersburg State University, 7/9 Universitetskaya Emb., 199034 St. Petersburg, Russia; 3Radcliffe Department of Medicine, University of Oxford, John Radcliffe Hospital, Oxford OX3 9DU, UK; credwood@well.ox.ac.uk

**Keywords:** congenital myopathy, disease-causing mutations, tropomyosin-troponin regulation, muscle contraction, spatial rearrangements

## Abstract

We have used the technique of polarized microfluorimetry to obtain new insight into the pathogenesis of skeletal muscle disease caused by the Gln^147^Pro substitution in β-tropomyosin (Tpm2.2). The spatial rearrangements of actin, myosin and tropomyosin in the single muscle fiber containing reconstituted thin filaments were studied during simulation of several stages of ATP hydrolysis cycle. The angular orientation of the fluorescence probes bound to tropomyosin was found to be changed by the substitution and was characteristic for a shift of tropomyosin strands closer to the inner actin domains. It was observed both in the absence and in the presence of troponin, Ca^2+^ and myosin heads at all simulated stages of the ATPase cycle. The mutant showed higher flexibility. Moreover, the Gln^147^Pro substitution disrupted the myosin-induced displacement of tropomyosin over actin. The irregular positioning of the mutant tropomyosin caused premature activation of actin monomers and a tendency to increase the number of myosin cross-bridges in a state of strong binding with actin at low Ca^2+^.

## 1. Introduction

Tropomyosin (Tpm) and troponin are key players in thin filament-based regulation of striated muscle contraction [1]. These regulatory components in coordination contribute to the work of actin-myosin complex triggering the force generation and, just as importantly, providing the relaxation. Via changes in actin, Tpm and troponin conformations, Tpm‒troponin is able to regulate the availability of the myosin-binding sites on actin in a Ca^2+^-dependent manner [2,3]. The transmission of Ca^2+^-activating signal to the elongated actin filaments is provided by Tpm dimers which bind each other in the head-to-tail manner, extend along and coils around actin. The dynamic displacement of Tpm from outer to inner actin domains is possible thanks to the electrostatic nature of the actin-Tpm interaction and flexibility of actin and Tpm [4,5]. Moreover, the bending stiffness and persistence lengths of actin and Tpm vary during force generation, causing an azimuthal shift of the tropomyosin strands [6]. It is postulated that Tpm occupies three equilibrium positions during contraction—the blocked, closed and open. At low Ca^2+^ concentrations, actin monomers are switched off and Tpm is in the blocked position where it covers the essential regions of myosin-binding sites on actin, anchoring over the outer actin domains in most energetically favorable position with roughly 30 electrostatic interactions with actin [7]. Elevation of cytoplasmic calcium concentration gives rise to a series of conformational rearrangements in the sarcomere. At first, the structure and conformation of troponin alters and some of the actin monomers change own conformation to the switched-on state. The persistence length of the actin filament decreases and that of Tpm strands increases [6], resulting in the Tpm movement to the closed position and in the partial unveiling of the binding sites for myosin on actin. The strong binding of myosin to actin moves Tpm deeper into the actin groove to the open position, most parts of actin monomers are switched-on and stimulate release of ADP and P_i_ from active site of myosin, significant conformational changes in the myosin head and generation of the power stroke. ATP binding to myosin changes the myosin head conformation, weakening the actin-myosin interaction and providing relaxation. Tpm takes its position over outer actin domains again, inhibits myosin binding with actin and the cycle resumes.

The structure of Tpm is well studied [8]. Almost throughout the entire length, this protein has a coiled-coil structure, which is formed by the repeating sequence of amino acid residues with non-polar (hydrophobic) and highly charged residues [9]. The hydrophobic amino acid residues pack into a hydrophobic central core. Highly charged residues on the Tpm surface form salt bridges between two chains of dimer that stabilize the molecule and also interact with the residues of the partner proteins.

Three genes TPM1, TPM2 and TPM3 code Tpm of skeletal muscles. The mutations in TPM2 and TPM3 genes lead to congenital myopathies—clinically and genetically heterogeneous muscle disorders that usually manifest at birth by muscle weakness and hypotension, slowly progressing over time [10]. The consequences of muscle dysfunction vary from mild scoliosis and delayed motor functions to difficulty in moving. Patients with severe form of disease may suffer from respiratory muscle weakness and respiratory failure, which requires regular monitoring from birth. The presence and progression of skeletal deformities can cause severe heart failure [11]. The most common causes of death are cardiopulmonary collapse, skeletal deformities and malignant hyperthermia. The range of skeletal muscle diseases with negative impact on quality of human life is quite wide. These are nemaline myopathy, cap myopathy, congenital fiber-type disproportion, actin myopathy, central core disease, intranuclear core myopathy, myotubular myopathy, centronuclear myopathy, distal myopathy and some other lesser-known diseases. The nomenclature of myopathies is still not approved and differs among different sources [12,13].

The correct diagnosis of congenital myopathy variant is critical for both the prediction of the disease course and decision making in the treatment strategy. In clinical practice, up-to-date methods of diagnosis and therapy suggest that genetic testing (NGS) follows the compilation of the clinical history of patient and the initial diagnostic workup [14]. Identification of genetic defect is required primarily due to the heterogeneity of clinical features. However, it has been found that genetic causes are also heterogeneous. A wide range of genetic defects might be responsible for the same basic phenotype. Moreover, the same gene mutation may be attributable to several different skeletal muscle diseases or to unspecified variant of myopathy [15,16,17,18]. Thus, NGS approaches should not be used in isolation, but in conjunction with traditional research techniques such as histopathology, biophysics and biochemistry in an integrated manner [14]. A biopsy of diseased skeletal muscles at congenital myopathy shows different histological changes including the presence of nemaline bodies, rods, cap structures, central cores, minicores, central nuclei and fiber-type disproportion and hypotrophy. Intracellular inclusions are distinguished by localization within the sarcomere and composition after staining by various methods [19]. The appearance of protein aggregates is most likely provoked by some unexplored compensatory processes in the muscle fiber due to the changes in the structural and functional properties of affected muscle proteins. The question, then, is to determine what different defects occur in thin filaments in order to cause the formation of various histological patterns. To address this question, it is important to elucidate the molecular mechanisms of impaired sarcomere contractility in the presence of the disease-causing mutations first of all in vitro.

The present study therefore was undertaken to determine the molecular mechanisms of the congenital myopathy caused by one of the mutations in TPM2, leading to Gln^147^Pro (Q147P) substitution of Tpm2.2. Originally reported as causing nemaline myopathy, the Q147P substitution was later clarified as cap myopathy according to the data of additional immunohistochemical and electron microscopic studies [20,21]. The clinical picture of the patient was relatively mild and differed from that of patients with the typical form of nemaline myopathy. Muscle biopsy showed predominance and selective hypotrophy of type I muscle fibers, numerous small muscle fibers with peripheral aggregates of rods in a cap-like fashion. Proline residue substitutes glutamic acid nearly in the middle of the Tpm molecule, on its surface. Proline does not embed in the normal structure of coiled-coil proteins because it forms a kink and can break a helix. The aim of the study was to evaluate the effect of the Q147P substitution on the spatial rearrangements of actin, Tpm and the myosin head at some simulated stages of the ATPase cycle. The polarized fluorescence from the probes bound to actin, Tpm or myosin subfragment-1 was registered in ghost fibers—the myosin-Tpm-troponin-free ghosts of skinned single fibers composed of more than 80% actin. Ghost fibers retain the overall highly ordered structure and therefore constitute a perfect model to study the conformational changes of actin and incorporated inside exogenous actin-binding proteins [22]. Thin filaments of ghost fibers were reconstructed with troponin and recombinant mutant or wild-type Tpms.

The objectives of the current study were to determine (i) the ability of the mutant Tpm to change its position over actin and to stimulate an increase or decrease in the number of activated actin-myosin cross-bridges at high and low Ca^2+^ concentrations; (ii) which targets can be used to alleviate the negative impact of the mutation and which potential drugs can be tested. We found the inability of troponin to switch the thin filaments off at low Ca^2+^ concentration and an increase in the number of the myosin cross-bridges strongly bound to actin. The angular orientation of the fluorescence probes bound to Tpm was found to be changed by the substitution and was characteristic for the shift of Tpm strands closer to the inner actin domains. It is proposed that the mutant Tpm can shift deeper into the actin groove and loses the ability to move on actin, which facilitates the binding of myosin cross-bridges to actin. The information about the therapeutic targets for the restoration of contractile function in the presence of the Tpm mutations is needed to initiate the testing of these compounds, primarily in model muscle fibers. To mitigate the damage evoked by the Q147P substitution, we propose to use small-molecule inhibitors of myosin, which can specifically suppress the functions of myosin-II—*N*-benzyl-p-toluenesulfonamide (BTS), butanedione monoxime (BDM) or blebbistatin [23].

## 2. Results

### 2.1. Analysis of Conformational Rearrangements of Actin and Tpm in Control

Using ghost fibers from rabbit psoas muscle, thin filaments were reconstructed sequentially with exogenous recombinant control (wild-type, WT) Tpm and troponin, and then modified by myosin subfragment-1 (S1). Three series of experiments were carried out, in each of which one of the proteins (actin, Tpm or S1) were modified with fluorescent probes. The polarized fluorescence intensities from actin-FITC, Tpm-AF and S1-AEDANS were registered and the angles of emission dipoles orientation *Φ_E_* of the probes, number of disordered fluorophores *N* (for S1-AEDANS) and bending stiffness (for actin-FITC and Tpm-AF) were calculated. Several functional states were simulated at low or high Ca^2+^ concentrations, in the absence or presence of MgADP and MgATP. Three-state model of striated muscle regulation [24] suggests that at low Ca^2+^ concentration troponin stabilizes Tpm strands closer to the periphery of thin filaments sterically blocking the myosin-binding sites on actin and inhibiting the interaction between these contractile proteins. However, in a native fiber it is impossible to block the binding of all myosin heads with actin and so the blocked state was simulated in the absence of myosin.

The changes in the *Φ_E_* value indicate spatial rearrangements of the protein or its extended region modified by the fluorescent probe. According to the data obtained (Figure 1a,b), in ghost fibers containing Tpm and troponin at low Ca^2+^ concentration the values of *Φ_E_* for the probes associated with actin (actin-FITC) and Tpm (Tpm-AF) are 46.2 degrees and 61.3 degrees, respectively. With elevated Ca^2+^ concentration in ghost fiber in the absence of the myosin heads, the *Φ_E_* values change and amount to 47.8 degrees (+1.6) for actin-FITC and 58.1 degrees (−3.2) for Tpm-AF. If S1 are introduced into fibers, and thereby induce the open functional state of thin filaments, then the *Φ_E_* values amount to 49.1 degrees (+1.3) for actin-FITC and 55.5 degrees (‒2.6) for Tpm-AF—the highest and lowest values in the histogram, respectively. Thus, as in the previous studies of conformational rearrangements of muscle proteins in ghost fibers [3,25,26], activation of thin filaments is accompanied by the increase in the *Φ_E_* value for actin-FITC and by the decrease in this value for Tpm-AF as compared to the simulation of the blocked state.

The FITC-phalloidin binds to the region of three neighboring actin monomers [27] and it can be assumed that an increase in the *Φ_E_* value indicates a rotation of actin monomers in the direction from the axis of the thin filament (clockwise). Taking into account current ideas about the regulation mechanisms related to actin, we can conclude that an increase in the *Φ_E_* value for actin-FITC is characteristic of actin monomers switching-on. The 5-IAF probe is associated with both cysteine Tpm residues in positions 36 and 190, and the changes in the *Φ_E_* value indicate the spatial rearrangements of the Tpm strands on actin. A decrease in the *Φ_E_* value for TM-AF was observed at simulation of activated state of thin filaments and is considered as being correlated with the azimuthal movement of Tpm towards the center of the actin filament (to the groove of the actin long helix) showed in electron microscopy works [7]. The Tpm shifting is possible due to an extension or shortening of the actin filament and the Tpm strands along the actin filaments. Based on this conclusion, the effects of the Q147P substitution in Tpm on the spatial rearrangements of actin and Tpm were analyzed below.

When actin monomers are activated, the bending stiffness value *ε* for actin decreases (Figure 2a); while on deactivation, on the contrary, it increases. The value of *ε* for Tpm varies ambiguously, but as a rule, it changes in the opposite direction as compared with actin (Figure 2b). As was proposed previously [7,28], the changes in bending stiffness of the filaments mean the changes in their persistence length. Usually, an increase in the persistence length of Tpm strands correlates with a decrease in the persistence length of actin filaments and vice versa. These changes easily explain the spatial rearrangement of Tpm relative to actin in the process of regulation of actin-myosin interaction [28].

In the competition between Tpm and myosin the latter wins, and myosin is able to bind with actin even at low Ca^2+^ concentrations. According to the data, the myosin heads activate actin and shift Tpm to a lesser extent as compared with a high Ca^2+^ concentration. The *Φ_E_* values do not reach their limit values and stop at 48.4 (−0.7) and 56.8 (+1.3) degrees for actin-FITC and Tpm-AF, respectively. It was expected that ADP would activate actin and change the Tpm localization even more, than in the nucleotide absence, however, this does not always work out.

Simulation of the strong-to-weak transition leads to a decrease in the *Φ_E_* value from 49.1 to 46.8 degrees (−2.3) for actin-FITC, and to an increase in the *Φ_E_* value from 55.5 to 58.4 degrees for Tpm-AF (+2.9). In the presence of ATP, actin reverts closer to the initial conformation—its monomers rotate towards the axis of the muscle fiber (anticlockwise). Such a decrease in the *Φ_E_* value is characteristic for the deactivation of actin filaments. The Tpm location also becomes closer to the initial position on actin, i.e., on the periphery of the actin filament, and such an increase in the *Φ_E_* value is characteristic for blocking the sites of myosin binding on actin by Tpm.

The *Φ_E_* values are slightly higher for actin-FITC and lower for Tpm-AF in the presence of ATP than the initial values (in the absence of S1) both at low and high concentrations of Ca^2+^. When using myosin heads modified with a fluorescent probe, we can still continue to register polarized fluorescence after addition of ATP for some time up to 2–3 h, although the fluorescence intensity decreases sharply. Consequently, ATP induces both the complete detachment of some myosin heads from actin, and the conversion of some of the myosin heads to a weak interaction with actin.

### 2.2. Analysis of Conformational Rearrangements of the Myosin Heads in Control

The fluorescent probe attaches to the myosin head in the region of SH1-helix in the immediate vicinity of the converter domain that change their orientation in ATPase cycle. We interpret the changes in the value of *Φ_E_* for S1-AEDANS (Figure 1c) as evidence of conformational rearrangements of the motor domain, leading to the tilt of the myosin head relative to the axis of the muscle fiber, or at least of the SH1-helix [3,29,30]. Regardless of this assumption, in our works there is a characteristic pattern of changes in the values of *Φ_E_* and *N* (number of disordered fraction) for the myosin heads—these values decrease during the formation of strong interaction with actin and increase during the formation of weak interaction. The value of *N* is related to the mobility of the probe and, therefore, to the affinity of myosin for actin (Figure 1d).

The minimal *Φ_E_* values 43.5 and 42.4 degrees is observed in the absence of nucleotides and in the presence of ADP and a high Ca^2+^ concentration. In these states, we observe a low mobility of the probe 0.298 and 0.328 rel. units, which means a high affinity of myosin for actin. At a low Ca^2+^ concentration and in the presence of ATP, the *Φ_E_* and *N* values increase. A strong increase in the *N* value to 0.589 rel. units in the presence of ATP can be explained by an increase in the number of freedoms of the myosin heads weakly associated with actin. Thus, the larger *Φ_E_* and *N* values characterize an increase in the number of heads weakly associated with actin; a smaller value of these parameters’ points to the myosin heads immobilization and their transition to a strong interaction with actin, involving a large number of interactions between S1 with both actin and Tpm. The characterization of changes in the *Φ_E_* and *N* values will be used further to analyze the Tpm mutation-causing changes in conformational rearrangements of the proteins.

### 2.3. The Influence of The Q147P Substitution in Tpm on Conformational Rearrangements of Actin, Tpm and the Myosin Heads

Using the data on polarized fluorescence of the probes associated with actin, Tpm and S1 in control, it is possible to analyze the effect of the Q147P substitution in Tpm on the conformational changes of the modified sarcomere proteins, and thereby to identify the primary defects which can lead to muscle dysfunction. For the reader’s convenience, changes in the values of *Φ_E_*, *ε*, and *N* caused by the presence of the mutant Tpm in muscle fiber relative to model fibers containing control Tpm were given.

Since the actin conformation determines the ATPase activity of myosin, it is important to monitor how the binding of the mutant Tpm affects it. An increase in the *Φ_E_* value for actin-FITC was characteristic for the switching actin monomers on, i.e., for their activation (Figure 1a), when they contribute to the activation of myosin ATPase [26]. From the data given in Figure 3a it follows that the mutant Tpm increases the *Φ_E_* value by 2.0 degrees at a low Ca^2+^ concentration in the absence of myosin heads and up to 1.4 degrees in the presence of strongly bound myosin heads (in the presence of ADP). The increase in *Φ_E_* parameter of polarized fluorescence for actin-FITC was typical for activation of actin filament in control. At a high Ca^2+^ concentration, the mutant, conversely, decreases the *Φ_E_* value by 0.6 degrees in the absence of S1 and by 0.7 degrees in the presence of strongly bound S1 (without nucleotide) that is typical for switching some of the actin monomers off. The *Φ_E_* increase correlates with a decrease in the bending stiffness *ε* (Figure 2a), that confirms the appearance of the additional amount of switched-on actin monomers.

The mutant Tpm-AF, when compared with the control, has the distinct parameters of polarized fluorescence (Figure 3b). The mutant decreases the *Φ_E_* value significantly by 1.7 to 6.7 degrees. The decrease in the *Φ_E_* value was typical for the spatial arrangement of the control Tpm in the presence of the strong-binding myosin heads. The bending stiffness ε of the mutant Tpm is significantly reduced by 1.6–3.5 degrees (Figure 2b) in all simulated states as compared with WT Tpm, so the mutant becomes highly flexible.

Myosin responds to the substitution in Tpm by a decrease in the values of *Φ_E_* at a low Ca^2+^ concentration by 0.8 degrees in the absence of nucleotide and in the presence of ADP. The rotation of probe towards the muscle fiber axis (*Φ_E_* decreases) and immobilization of the probe (N decreases) for S1-AEDANS is typical for the transition of a larger number of heads to a strong-binding form with actin.

The simulation of weak binding of the myosin heads with actin was performed with addition of 3 mM MgATP to the muscle fiber. In the presence of ATP, the mutant Tpm decreases the *Φ_E_* values for actin-FITC by 0.6–0.8 degrees and for Tpm-AF by 1.7–1.9 degrees, and increases the values of *Φ_E_* for S1-AEDANS by 1.1–5.0 degrees at high and low Ca^2+^.

In control, the addition of ATP was accompanied by the decrease in the *Φ_E_* value for actin-FITC and by the increase in this value for S1-AEDANS and Tpm-AF as compared with the absence of ATP (Figure 1). Tpm should return in an energetically favorable position over outer actin domains and actin monomers should be deactivated. It means that actin adopts the conformation that cannot activate ATP hydrolysis in myosin active site. The weak interaction between actin and myosin suggests the decrease in the number of bonds between these two proteins, that was schematically presented in Figure 4. The changes in polarized parameters for the mutant Tpm-AF in the presence of ATP (decrease in the values of *Φ_E_*) reveal the aberrant positioning of the mutant Tpm. The decrease in *Φ_E_* parameter of polarized fluorescence for actin-FITC was typical for deactivation of actin filaments. Therefore, actin is deactivated in the presence of the mutant Tpm and even more actin monomers transit to the switched-off conformation. Increase in the *Φ_E_* value for S1-AEDANS indicates on the transition of the myosin heads to the weak-binding conformational state. Thus, the relaxation process occurs and are not significantly affected by the substitution in Tpm.

Therefore, the principal effect of the Q147P substitution covers the stages of the cross-bridges cycle that occur before the force generation at a low Ca^2+^ concentration. The coincident pattern of the changes in the parameters for actin-FITC, Tpm-AF and S1-AEDANS at a low Ca^2+^ concentration show the abnormal premature activation of contractile system in the presence of the mutation.

## 3. Discussion

The structural and functional effects of a number of Tpm mutations associated with different variants of congenital myopathy have been investigated previously [29,31,32]. It was found that the Tpm mutations can change several features of Tpm and affect different aspects of muscle regulation. One of the most important features affected by Tpm mutations is a Ca^2+^-sensitivity of thin filaments activation which can be caused directly if a mutation occurs in the troponin-binding site of Tpm or indirectly through the changes in cooperativity. The mutations can both increase or decrease or not change thin filament Ca^2+^-sensitivity [33]. In the presence of the Tpm mutations, the equilibrium between the formation of different functional states of the thin filament can shift toward the blocked or open state, and thus the pool of myosin heads, ready to form a strong binding to actin and generate force, changes [34]. The mutations that occur in Tpm genes can both locally and globally alter the structure or conformation of Tpm and its surface charge. The replacements of a hydrophobic residue (for example, Ala^155^Thr) can damage the hydrophobic core, change the bend of the Tpm strands [35] and cause a violation in the Tpm position relative to actin and myosin-binding sites on actin [25]. The mutations resulting in charged to neutral or oppositely charged radical residue substitution (e.g., Glu^41^Lys, Arg^91^Gly, Glu^117^Lys), can modify the energy landscape of Tpm [34], and the electrostatic interaction of Tpm with actin or troponin (and possibly also with myosin) may weaken or intensify [36]. Even local changes in Tpm structure are able to change the pattern of the protein-protein interaction in the complex of Actin‒Myosin‒Tpm‒Troponin and to disrupt the precise mechanisms of the actin-myosin motor functioning [32,37,38,39,40]. In addition to the disturbance of the actin‒Tpm‒troponin relationship, a change in the functioning of some other proteins that interplays with Tpm and actin, in particular, tropomodulin, nebulin, leiomodin, cofilin may also occur [41,42,43,44,45]. A serious consequence of this disruption may be muscle weakness and the development of myopathy in human and animals. It goes without saying that knowledge on the primary disorders taking place in the sarcomere due to mutations is necessary for achieving an effective approach to treat distinct variants of skeletal myopathies.

A number of Tpm mutations are able to reduce the affinity for actin. The Q147P substitution investigated in the present study dramatically reduces the affinity of Tpm for actin in co-sedimentation assay [32,37,46]. However, the reduced affinity for actin does not seem to equate to reduced incorporation in vivo [43,47], with normal [48] and abnormal [15] distribution of the different mutant Tpms in thin filaments being reported. The Tpm polymerization on actin in the fiber is a highly cooperative process. During early stages of thin filament assembly, the weak binding of Tpm is expected. At filament maturation and formation of the Tpm continuous cable, gaps are eliminated and Tpm binds to actin much more strongly. It turned out that even though the affinity of a mutant Tpm for actin reduces dramatically in solution, as was shown for the Q147P (6–7 times) [32], the content of the mutant Tpm associated with actin in a well-structured system of muscle fiber differs from the control not so significantly (1.7 times for the Q147P Tpm [30]). Apparently, during the thin filament assembly the mutant tropomyosin remains bound to thin filaments in the muscle fiber due to the end-to-end associations, wrapping around the F-actin many times and formation of a continuous cable. The ability of Tpm to bind with actin in sarcomere and functioning as a poison protein was also confirmed by the successful in vitro motility assay (IVMA) studies, with use of increased Tpm concentration [46]. The Q147P substitution was found to significantly reduce the maximal sliding velocity of reconstituted thin filaments in IVMA measured at saturating Ca^2+^ concentrations, and slightly decrease the Ca^2+^ sensitivity [46].

In the previous study of the Q147P effects done in the absence of troponin [30] we revealed the mutant Tpm shift towards the inner actin domains, the increase in the number of the myosin heads in a strong-binding form with actin, and the activation of an additional amount of actin monomers. However, the effect of troponin and Ca^2+^ on the molecular mechanisms of the regulation of actin-myosin interaction was not investigated. An effort was made in the present study to know how the Q147P substitution affects Ca^2+^-dependent functioning of muscle proteins.

The present study was carried out in the ghost fibers with regulated thin filaments reconstructed from exogenous Tpm and troponin. According to the polarized fluorescence measurements, the Q147P substitution changes the spatial rearrangements of actin filaments, Tpm and the myosin heads during ATPase cycle. The advantage of such approach is that the results can be obtained directly in a highly ordered system of a muscle fiber with high sensitivity under different experimental conditions. To analyze the spatial rearrangements, some fluorescence parameters were examined: the values of *Φ_E_* (the angles of emission dipoles orientation of the probes), *ε* (bending stiffness) and *N* (number of disordered fractions of the probes). The experiments were performed in parallel with control fiber system containing wild-type Tpm (without substitution). Therefore, all changes in the values of *Φ_E_*, *ε*, and *N* caused by the substitution are presented relative to the control (Figure 3 and Figure 4). In control, the activation of thin filaments by troponin-high Ca^2+^ and the myosin heads is accompanied by the increase in the *Φ_E_* value for actin-FITC, and by the decrease in this value for Tpm-AF and S1-AEDANS. This pattern of changes has also been observed in our previous studies [3,26,28]. The changes in the value of *Φ_E_* for Tpm-AF relative to actin are interpreted here according to a widespread theory of the Tpm shifting over actin surface from the outer to the inner actin domains [4]. High-resolution structure of the actin-Tpm complex in the different functional states of thin filament proposes the azimuthal movement of Tpm upon transitions between the states [5]. The polarized fluorescence technique can detect the angular changes of the probe orientation, but not their radial shift [49,50]. However, we found [3,26,29] that the changes in the value of *Φ_E_* for Tpm-AF correlate with the movement of Tpm observed by EM studies [5,7,51]. Since the decrease in the value of *Φ_E_* for Tpm-AF is observed in the control under activation conditions, then such changes are considered by us as a shift of Tpm strands towards the inner actin domains [3,26,29]. The change in a position of the Tpm strands relative to the inner domain of actin may be associated with a disparity in the alterations in the persistence length of Tpm strands and actin filaments that presumably cause azimuthal shifting of the Tpm strands [6,26,28,29]. For example, if the Tpm strands at transition from the ON to the OFF state undergo a greater elongation than does F-actin, it may cause an azimuthal shift of the Tpm strands towards the outer domain of actin [6,26,28,29,51]. Conversely, a lower compared to F-actin shortening of the Tpm strands move them to the inner domain of actin [6,26,28,29,51].

According to the changes in the value of *Φ_E_* for Tpm-AF, the mutant Tpm’s spatial orientation is characteristic of a shift from periphery of actin filament to the inner actin domains in all observed states where it should expose the myosin-binding sites on actin. The mutant-induced changes in the *Φ_E_* values for actin-FITC (Figure 3) mean that an additional amount of actin monomers is activated at a low Ca^2+^ concentration in the absence of myosin heads, as well as in the presence of strongly bound myosin heads. However, at a high Ca^2+^ concentration, the population of switched-on actin monomers does not reach the normal level.

It can be assumed that the number of bonds between actin and the mutant Tpm is lower compared to the control, that is confirmed by a decrease in the amount of Tpm bound in the compartment of actin filaments [30] and by the localization of the mutant Tpm. The increase in the flexibility of the mutant Tpm means that the persistence length decreases in the presence of the mutation. Moreover, the change between values of *ε* at high and low Ca^2+^ concentrations becomes subtle and often insignificant, that agree with the inhibited spatial rearrangement of Tpm over actin filaments at ATPase cycle (Figure 4). The bending stiffness of actin at a low Ca^2+^ concentration is significantly decreased by the mutation in two simulated states: in the absence of S1 and in the presence of S1 without nucleotide (Figure 2a), that confirmed the appearance of the additional amount of switched-on actin monomers. The premature activation at a low Ca^2+^ of thin filaments containing the mutant Tpm induce transition of a larger number of myosin heads to a strong-binding form (*Φ_E_* decreases, *N* decreases). It is at the stage when Tpm should normally prevent the strong interaction of myosin with actin, an abnormal activation of F-actin by the myosin heads occurs and a part of myosin passes into a strong binding with actin.

Since the substitution Q147P in Tpm weakens the ability to bind well with actin, one can expect the loss of a certain number of tropomyosin molecules and an increase in gaps. It is well known that in the presence of Tpm and in the absence of troponin the relative amount of switched-on actin monomers and the myosin heads strongly associated with actin increases. It seems possible that if the mutant Tpm left the muscle fiber area, the amount of switched-on actin monomers and the myosin heads in strongly-binding conformation would most likely decrease rather than increase. However, in our experiments performed in the absence of troponin [30] in the presence of the Q147P-mutant Tpm the relative amount of activated actin monomers and the myosin heads in strongly-binding conformation was even greater than in the presence of the WT Tpm. Thus, it can be considered that the activation of thin filaments in the presence of the mutant Tpm is not associated with a decrease in the affinity of the mutant Tpm to actin, but is most likely due to the abnormal behavior of Tpm on thin filaments and to the specific response of actomyosin to this disturbance.

Though the mutant Tpm strands are localized closer to the inner actin domains, they are not able to prevent actin deactivation in the presence of ATP and the myosin heads. A greater number of the myosin heads, following an additional actin deactivation, proceeds to a weak interaction with actin, as evidenced by a significant increase in the values of *Φ_E_* and *N* for S1-AEDANS. Thus, the relaxation process occurs. The mutant Tpm stimulates the transition of the conformational state of actin-myosin complex to the weak-binding form.

Therefore, the data obtained in two different muscle fiber models (in the absence of troponin [30] and in the presence of troponin) are consistent with each other and confirm that the main effect of the Q147P substitution is an abnormal Tpm positioning relative to the myosin-binding sites on actin, a failure of the Tpm ability to block the interaction of myosin with actin (first of all at a low Ca^2+^), and an increase in the number of strong-binding myosin heads. The number of strong-binding myosin heads decreases at a high Ca^2+^ concentration (Figure 3c). Such a change correlates with a decrease in the ATPase activity observed previously IVMA [46]. However, when ATP was added to muscle fiber without troponin, the myosin heads remained in a strong binding with actin. Troponin allows relaxation of the actin-myosin system in muscle fiber in the presence of the mutant Tpm.

Similar defects in Tpm inhibiting function were shown by us earlier for two other Tpm mutations. The first leads to a deletion of Glu^139^ in Tpm2.2 (E139X), identified in cap disease [47], as well as in a family with two siblings affected by congenital myopathy showing both nemaline rods and cap-like structures [52]. The second mutation causes the replacement Ala^155^Thr (A155T) in Tpm3.12 detected in a patient with nemaline myopathy, as well as in a patient with an undetermined variant of myopathy [53]. 80% of the fibers contained classical nemaline bodies. Muscle weakness with these mutations appeared early, and all patients had breathing difficulties. Type 1-fiber predominance and fiber-type disproportion were also found. A clinical and histological overlap between nemaline and cap myopathies was noticed earlier [54]. Some authors consider cap myopathy as a subtype of nemaline myopathy [12,13,54]. However, the localization and composition of nemaline bodies and caps are significantly different, and various mechanisms of their appearance can be proposed. Both nemaline bodies and cap structures simultaneously in the sarcomere can be detected [55]. The patient with the Q147P substitution had a mild form of myopathy, but his clinical picture was different from that of patients with the typical form of nemaline myopathy [20]. The peripheral aggregates of rods in biopsy, originally classified as nemaline bodies, were later considered as cap structures. Nemaline bodies are derivatives of the Z-discs and thin filaments; caps are peripheral regions of disorganized thin filaments. Actin, myotilin, nebulin, tropomyosin is found in both structures [54,56]. Desmin clustered aggregates, troponin T, and SERCA in caps and γ-filamin, cofilin-2, telethonin, and α-actinin in nemaline bodies were identified. Probably, caps and nemaline bodies formation have similar mechanisms, through disruption of Z discs and thin filaments. It is possible that there are also other factors that determine the consequence of the malfunctioning of the actin-myosin system due to the inclusion of the mutant Tpm in the structure of thin filaments and various histological anomalies. The question on the formation of cap and nemaline bodies mechanisms remains open.

The first thing we noticed is the close proximity of all these mutations, the disruption in the coiled-coil structure induced by them and the decrease in the mutant Tpm affinity for actin. The E139X deletion and the Q147P and A155T substitutions occur in the central region of Tpm dimer whose flexibility is essential for the proper functioning of the protein and primarily for actin binding [57]. Tpm has an almost perfect coiled-coil structure and its heptad repeats do not interrupt throughout its sequence. Despite this property, there are several destabilizing clusters in the Tpm molecule [58]. The least stable is the central region due to the presence in the hydrophobic core a polar Asp^137^ residue non-canonical for the coiled coil that produces a hole between two α-helices of Tpm. Asp^137^ is followed by the fourth alanine cluster Ala^151^-Ala^155^-Ala^158^, which gives the Tpm molecule a small radius at this region and a bend of coiled coil. Apparently, all these structural features together make a large contribution to the binding of Tpm with actin.

The E139X deletion and the Q147P and A155T substitutions reduce the affinity of Tpm binding for actin in co-sedimentation assay [30,32,37]. The affinity of the E139X Tpm for actin reduces by about 6 times, just like Q147P [32]. The deletion of Glu^139^ leads to a disruption of direct interaction of Tpm with Lys^326^ residue of actin [59] and to the local violation in the coiled-coil region 138–154 [56]. Replacement of Gln^147^ by helix-breaker proline also creates the interruption of the Tpm coiled coil in the region of 144–154 residues [56]. The Ala^155^ residue does not show a participation in the direct interaction with actin, but replacing it with Thr leads to a decrease in the Tpm affinity for actin by 2.5 times [37]. Probably, such a decrease occurs due to increased Tpm rigidity [60] and restriction of the mutant bending and adopting the appropriate conformation [37]. Despite such a strong decrease in affinity for actin, the E139X, Q147P and A155T Tpms incorporate into sarcomeres [26,30,47,60]. Using confocal microscopy and electrophoretic separation of muscle fiber proteins, we showed that all three mutants can be integrated into the compartment of thin filaments of a single muscle fiber. The content of the Tpms bound to actin decreases by 40%, 15% and 4% in the presence of the Q147P, E139X, and A155T mutations, respectively. According to the polarized fluorescence data, all three mutations decrease the threshold of the Ca^2+^-sensitive activation in the spatial rearrangements of actin, the myosin head and Tpm. Even at low Ca^2+^, activation of a certain number of the myosin heads and actin monomers and localization of Tpm closer to the inner actin domains were proposed. As previously reported, the affinity of Tpm for actin is reduced in the gain-of-function mutations, and the energetically most favored position of such mutant Tpms is shifted in the direction of the open state [59]. Our data confirm this idea.

Considering that the β-isoform of tropomyosin (Tpm2.2) as a homodimer is most likely nearly absent in muscle and is present as αβ-heterodimer, reduced Tpm affinity for actin in vivo can be compensated additionally by the presence of α-isoform without mutations in the Tpm coiled coil, and the incorporated mutant can work as a poisonous protein. On the basis of our previous study on the effect of the Arg^133^Trp (R133W) substitution in the β-chain of ββ-homodimer and αβ-heterodimer [61] we suggest, that the investigation of the mutation effect using homodimers is acceptable. We believe that the αβ-heterodimer is a more appropriate model for studying the effects of myopathic mutations in the β-chain of Tpm. The negative effects of the mutation can be partially compensated for by the second tropomyosin chain, which does not contain the mutation. Despite the quantitative differences in the effects, the qualitative nature of those critical changes in the conformational state of actin, myosin, and Tpm, which occurred as a result of the amino acid substitution R133W, was the same for the αβ-heterodimer and ββ-homodimer of Tpm [61]. The key aspects of impaired contraction regulation were manifested much more clearly with study of homodimers with mutations in both subunits. Thus, the results give an important information about the molecular mechanisms of congenital myopathy associated with the Q147P substitution in Tpm and, we believe, will help for the testing of potential drugs to eliminate the primary causes of muscle dysfunction.

In conclusion, we propose that congenital myopathy-causing substitution Q147P in Tpm2.2 makes it more difficult for Tpm strands to keep in the blocked position and easier for them to shift over the inner actin domains, towards the actin groove. The flexibility of the mutant Tpm increases. The altered position of Tpm on the thin filament prematurely activates actin monomers and stimulates myosin to form strong binding with actin even at a low Ca^2+^-concentration. The relaxation state of contractile system does not suffer due to the Q147P substitution. It is important to compensate for the premature activation of the actin-myosin interaction caused by the presence of the Q147P mutant tropomyosin. To mitigate the damage evoked by the Q147P substitution, we propose to use small-molecule inhibitors of myosin that can specifically suppress the functions of myosin-II. The testing of the myosin inhibitor is the objective of the next stage of our research.

## 4. Materials and Methods

### 4.1. Proteins Preparation and Its Modification by Fluorescent Probes

In the study, recombinant preparations of the control (wild-type) and mutant human tropomyosins Tpm2.2 were used. Protein expression was carried out using the bacterial culture BL21 (DE3) pLysS *Escherichia coli* (Novagen, Madison, WI, USA) [62]. The Tpm coding sequence contained codons for the inclusion of two additional Ala-Ser N-amino acid residues in the protein to compensate for the reduced affinity of non-acetylated recombinant Tpms for actin [63].

Myosin was isolated from rabbit skeletal muscles [64] and digested by α-chymotrypsin (Sigma-Aldrich, St. Louis, MO, USA) to subfragment-1 (S1) [65]. Troponin was obtained from acetone powder prepared from rabbit skeletal muscle [66] and then purified by chromatography using a BioLogical LP System low-pressure chromatograph (BioRad, Hercules, CA, USA). The preparations purity was analyzed by vertical polyacrylamide gel electrophoresis in the presence of sodium dodecyl sulfate using a Mini-PROTEAN^®^ Tetra camera (BioRad, Hercules, CA, USA). Intact muscle fiber F-actin, S1, and Tpms were specifically modified with FITC-phalloidin, 1,5-IAEDANS, and 5-IAF fluorescent probes (Molecular Probes, Eugene, OR, USA), respectively, in three different series of experiments. FITC-phalloidin molecules are located in the groove of the actin helix formed by actin monomers and specifically bind to three neighboring actin monomers [27]. The fluorescence data from FITC-actin helped us to obtain information on the spatial organization of actin monomers in the F-actin helix and actin filament flexibility [60,67]. Myosin heads (myosin subfragment-1) was selectively modified at Cys^707^ with 1,5-IAEDANS fluorescent probe in order to obtain information on the change in the character of the myosin binding to actin, in particular, on the relative number of cross-bridges strongly associated with F-actin in muscle fiber and their mobility [3]. The relative localization of Tpm and the flexibility of its strands in the thin filaments of muscle fiber was determined using 5-IAF fluorescent probes linked to Cys^36^ and Cys^190^ [68], which provide information from the entire protein molecule [3].

### 4.2. Ghost Muscle Fibers Preparation

Rabbit psoas muscle was isolated to prepare model fibers [69]. The animals were killed in accordance with the official guidelines and regulations of the community council on the use of laboratory animals. All procedures were approved by the Animal Ethics Committee of the Institute of Cytology of the Russian Academy of Science (Assurance Identification number F18-00380, period of validity 12.10.2017–31.10.2022). Bundles of muscle fibers 3–4 cm in length were placed in a glycerinating solution. Before the experiment, single fibers were isolated from a bundle and placed in an extraction solution (K, Na-phosphate buffer with high ionic strength, containing 10 mM ATP, Sigma-Aldrich, St. Louis, MO, USA) [22]. The extraction of myosin, troponin and tropomyosin resulted in ghost fibers consisting of 70%–80% pure actin filaments and Z-lines proteins.

Isolated beforehand proteins (Tpm, troponin and S1) were incorporated into ghost fibers by incubation of them in the solution containing 2 mg/mL of the protein. The final concentrations of WT Tpm, troponin and S1 in modelled fibers was determined in separate experiments previously [3,26]. The experiments conditions did not change. The incubation time of 2–3 h was enough to achieve the molar ratio of the control Tpm and troponin to actin to be 1:6.5 (±2) and 1:6.5 (±2). Moreover, in the presence of the regulatory proteins, the molar ratio of S1 to actin was determined to be 1:5 (±2) in the absence of the nucleotides, 1:4.5 (±2) in the presence of ADP and 1:14 (±2) in the presence of ATP, respectively. Thus, the regulatory and contractile systems of the required protein composition were restored. Reconstructed ghost muscle fibers are excellent models for studying the conformational rearrangements of incorporated proteins modified by a fluorescent probe [3] and for testing various chemical compounds.

### 4.3. Polarized Microfluorimetry Technique

Using a polarized microfluorimeter, we measured the fluorescence characteristics of probes bound to actin, myosin head and Tpm strands. Changes in the spatial location and flexibility of probes were interpreted as conformational rearrangements of the proteins or extended probe area. The polarized fluorescence was recorded at 500–600 nm after excitation by a mercury lamp DRSH-250 at 407 ± 5 nm from 1,5-IAEDANS-labeled S1, 489 ± 5 nm from 5-IAF-labeled Tpm and FITC-phalloidin-labeled actin. The measurements were obtained in the presence of the WT and mutant Tpms incorporated into different muscle fibers in the parallel experiments. In the presence of S1, which is known to increase the Tpm affinity to actin [70], the sum of fluorescence intensities for Tpm-AF (I_sum_ = (_⊥_I_⊥_ + _||_I_||_ + _||_I_⊥_ + _⊥_I_||_) were similar for both Tpms: 1034 ± 23 for WT Tpm, and 978 ± 11 for the mutant Tpm. The activation and relaxation states of the contractile system and various functional states of thin filaments were modeled. Muscle fibers were incubated in various solutions containing 4 mM EGTA (Sigma-Aldrich, St. Louis, MO, USA) to remove free Ca^2+^ from the sarcomere, 0.1 mM CaCl_2_ to simulate the Ca^2+^ concentration increase to 10^−5^ M, 3 mM MgADP (and also in the absence of nucleotides) for the formation of strong-binding conformational state of myosin and 3 mM MgATP for the formation of weak-binding state. To analyze each conformational state of the protein complex, 25–30 measurements of fluorescence intensities were performed on 5–6 ghost fibers. Model-dependent [71] and model-independent methods [50] were used to determine values of *Φ_E_*—the orientation angles of emission dipoles of fluorescent probes, *N*—the number of randomly oriented dipoles and *ε*—the bending stiffness [72,73,74]. The model is based on the assumptions described in detail earlier [3]. According to the theory of a semiflexible filament, the angle θ between the fiber axis and thin filament is not zero and sin^2^θ = 0.87(kT/*ε*) L. The bending stiffness *ε* for actin and Tpm filaments was estimated from sin^2^θ [74]. According to the previous observations [75], an increase in the *Φ_E_* value for FITC-actin indicates an increase in the relative amount of switched-on actin monomers. A decrease in the *Φ_E_* value for 5-IAF-Tpm and 1,5-IAEDANS-S1 occur when the myosin heads transit to a strong binding with actin, and Tpm shifts to the groove of actin (towards the inner domains of actin). A change in the *N* parameter for 1,5-IAEDANS-labeled S1 correlates with the affinity of S1 for actin and the formation of a strong or weak binding. The parameter *ε* fluctuates with the flexibility of actin filaments and Tpm strands.

The effect of the change in the amount of the WT Tpm in thin filaments of ghost fiber has been investigated when developing experimental protocols. A decrease in the time of incubation of ghost fiber in a solution containing WT Tpm induced a decrease in the Tpm concentration in the thin filaments of ghost fiber. By this way, the activation and inhibition effects of the WT Tpm at a low and a high Ca^2+^ on actin-FITC and S1-AEDANS were compared to know how the concentration of Tpm can affect the polarized fluorescence parameters. We did not observe the different effects of Tpm on the polarized fluorescence parameters in dependance on the saturation of thin filaments by Tpm in the range of Tpm-actin molar ratio 1:3.5–1:6.5 (±2). At a high Ca^2+^, we registered the changes in parameters of actin-FITC and S1-AEDANS that were typical for activation of actin and for an increase in the pool of S1 strongly bound with actin. At a low Ca^2+^, the changes in parameters of actin-FITC and S1-AEDANS were typical for switching actin off and for a decrease in the pool of S1 strongly bound with actin. However, the quantitative effect was weaker than in ghost fibers with thin filaments saturated with Tpm.

## Figures and Tables

**Figure 1 ijms-21-07590-f001:**
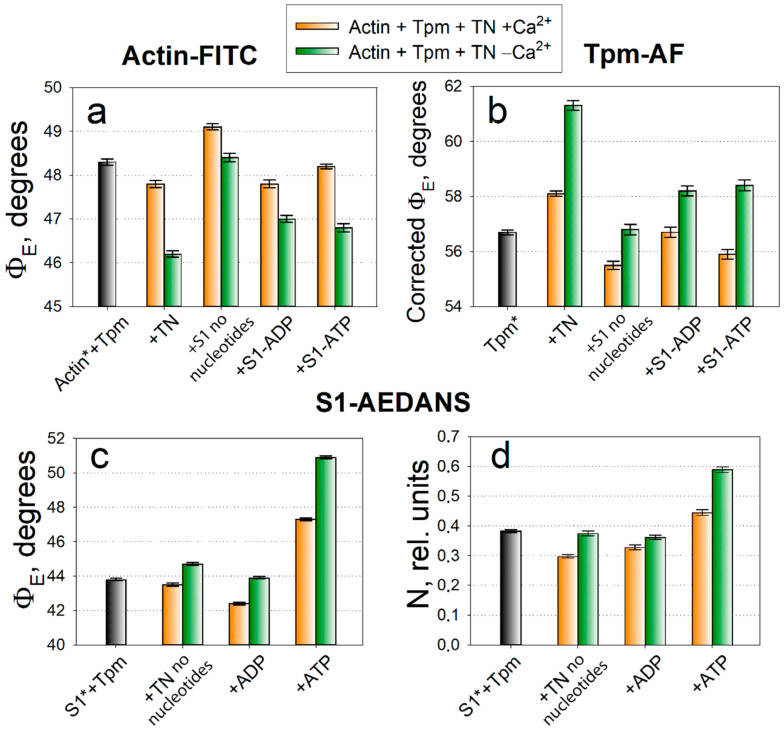
A typical pattern of changes in the angle of orientation of emission dipoles *Φ_E_* for fluorescent probes associated with actin (Actin-FITC, or Actin*, **a**), tropomyosin (Tpm-AF, or Tpm*, **b**) and in the values of *Φ_E_* and the relative number of disordered fluorophores *N* for probe bound to S1 (S1-AEADANS, S1*, **c**,**d**) in ghost fibers containing reconstituted thin filaments with control recombinant Tpm (of wild type, WT) and troponin (TN). The values are compared at high (orange columns) and low (green columns) Ca^2+^ concentrations. Black color is used for the values obtained in ghost fibers not containing troponin.

**Figure 2 ijms-21-07590-f002:**
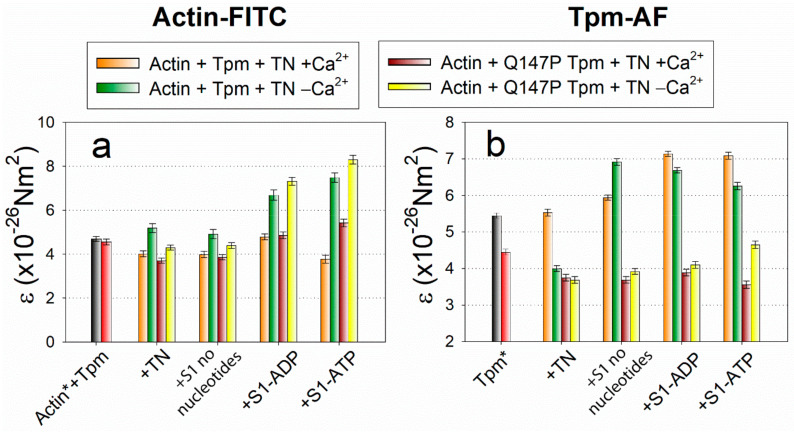
The changes in the value of *ε*—bending stiffness—for the fluorescent probes bound to actin (Actin-FITC, or Actin*, **a**) and Tpm (Tpm-AF, or Tpm*, **b**) in ghost fibers containing recombinant WT and Q147P-mutant Tpms in the absence of troponin (black and red columns) and in the presence of troponin at low and high Ca^2+^-concentrations (see legend). The increase in the value of *ε* means the increase in the persistence lengths of actin and Tpm filaments.

**Figure 3 ijms-21-07590-f003:**
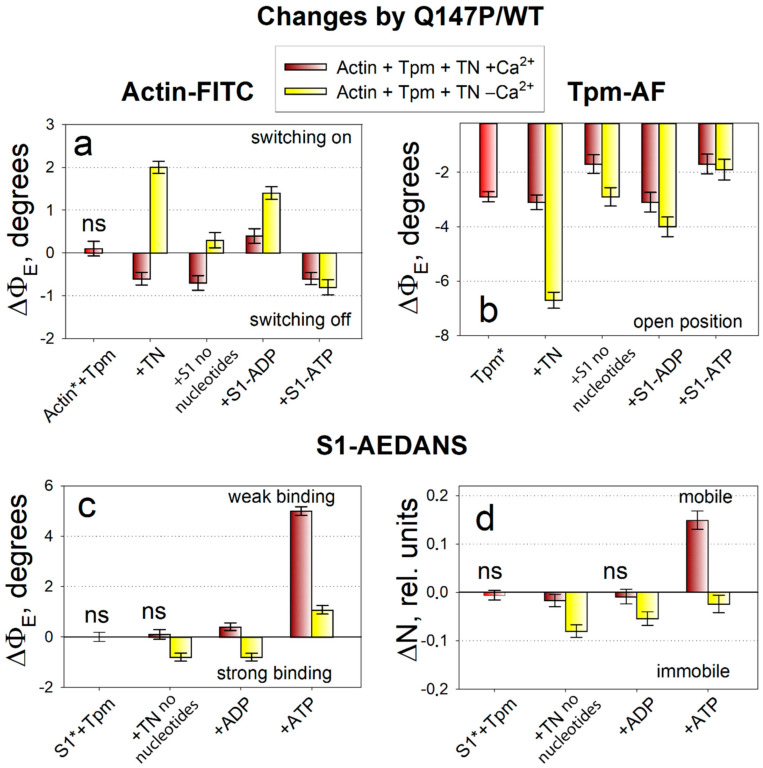
The effect of the Q147P substitution in Tpm on the angles of orientation of emission dipoles *Φ_E_* for fluorescent probes associated with actin (Actin-FITC, or Actin*, **a**), tropomyosin (Tpm-AF, or Tpm*, **b**) and on the values of *Φ_E_* and relative number of disordered fluorophores *N* for fluorescent probe associated with S1 (S1-AEADANS, or S1*, **c**,**d**) in ghost fibers containing reconstituted thin filaments with mutant recombinant Tpm and troponin (TN) at high (dark red columns) and low (yellow columns) Ca^2+^ concentrations. The changes in the values are given relative to the WT Tpm (see Figure 1) in the same order and are significantly different (p < 0.05) except the values indicated as nonsignificant—ns. Red color is used for the values obtained in ghost fibers not containing troponin. An increase in the value of *Φ_E_* for actin-FITC occurs at switching actin on, while a decrease in the values of *Φ_E_* for Tpm-AF and S1-AEDANS is typical for the shift of Tpm towards the open position in actin groove and for increase in the number of the myosin heads in the strong-binding state.

**Figure 4 ijms-21-07590-f004:**
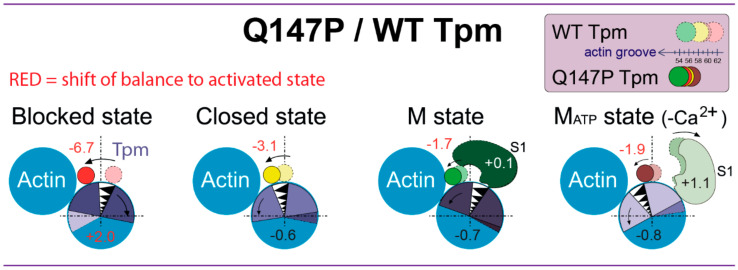
The schematic view of the changes in the conformational rearrangements of actin monomers and the myosin heads and in the position of Tpm caused by the congenital-myopathy-causing mutation Q147P in Tpm2.2. The muscle fibers contained Tpm and troponin at a low Ca^2+^ concentration (simulated blocked state), at a high Ca^2+^ concentration (closed state) or Tpm, troponin and S1 in the absence of nucleotide (M state) and in the presence of ATP (M_ATP_ state). Several proposed positions of Tpms related to the values of *Φ_E_* are shown on top. The numbers close to Tpm, actin, and the myosin head are used to designate the changes under the substitution in the values of *Φ_E_* for actin-FITC, Tpm-AF and S1-AEDANS as compared with the respective changes in the presence of WT Tpm.

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
