# Peer review of "Looking for Targets to Restore the Contractile Function in Congenital Myopathy Caused by Gln147Pro Tropomyosin"

_ijms, 2020, doi:10.3390/ijms21207590_

Round 1

Reviewer 1 Report

The Karpicheva et al. investigated a congenital myopathy caused by mutated Glu147Pro of TPM2 in β-tropomyosin. They observed the spatial rearrangements of actin, myosin, and tropomyosin in the single muscle fibre during the simulation of several stages of ATP hydrolysis cycle. As a result, the Glu147Pro mutation decreases the number of bonds between tropomyosin and actin and disrupts the myosin-induced displacement of tropomyosin over actin. They concluded that the negative impact of this TPM2 mutation may be reversed by myosin inhibitors.

Overall, this article is interesting and adds novel findings to the current knowledge about the complex underlying pathomechanisms of a diverse spectrum of congenital myopathies. However, there are still several issues needed to be clarified and not a few grammatic errors of the current form of the manuscript.  

The major criticism for this manuscript:

  1. The length of the introduction section should be reduced; otherwise, a large portion of the content should be moved to the Discussion Section.
  2. The “ Results and Discussion” section should be separated as Results and Discussion for a better understanding for the readers.
  3. I think in the Introduction Section, the authors should first give a short introduction regarding what did the “ghost fibre” mean? and how did it represent in this study? Should it be more appropriate to be called “modeled“ or “controlled” muscle fibre?
  4. The authors proposed that the negative impact of the mutation may be alleviated with the use of myosin inhibitors. However, I have seen any evidence provide in the current study which supports their postulation. Did the author apply any compound of myosin inhibitor in treating their cells? I suppose that the manipulation of Ca2+ would be more straight-forward to affect this negative impact of the mutation.

The minor criticisms for this manuscript:

  1. The presentation of Glu147Pro should be "Glu147Pro."
  2. Page 2, Line 90-96: In the era of genetic diagnosis, it has been suggested to do genetic testing (NGS) directly after consulting patients with suspected congenital myopathies, and to put images and invasive muscle biopsy behind the supporting evidence. It seems that the progress of the times can make neuromuscular patients get more invasive tests with less information. Therefore, I think the paragraph should be rewritten to more precisely describe the updated diagnostic algorithm for congenital myopathies, especially the less dependent on the basis of histological changes noted in the biopsy of skeletal muscles. I suggest the authors may refer to this updated article in Lancet Neurol. 2020 19: 522-532.
  3. Page 2, Line 83: The description of “even heart failure” is incorrect. This sentence should be rewritten.
  4. Page 2, Line 91: The term "skeletal myopathies" is inappropriate. Should it be congenital myopathies, muscular dystrophies, or metabolic myopathies?
  5. Page 3, Line 109 "Congenital myopathy can be regarded as a disease of a thin filament" This description of congenital myopathy is not appropriate, the authors should rewrite it and give a more comprehensive view of the pathomechanism of congenital myopathies.
  6. Page 3, Line 130: Cap myopathy can be caused by mutations in the ACTA1, TPM2, or TPM3 genes. The Q147P mutation belongs to the TPM2 category.
  7. Page 4 Line 156: Given the fundamental effect on myosin, the authors should more detailedly explain the mechanism of "cross-bridges " and its effect. Does it mean "Tpm shifts into the actin groove and ability to move on actin"?
  8. Page 4, Line 166: “decorated” should be corrected as “modified”.
  9. Page 4, Line 172-174: The sentence of "The angles of emission dipoles ..., respectively” is not described correctly and somehow confused to readers, and it should be rewritten for better understanding.
  10. Page 5, Line 190-191: The sentence of "The changes in the ΦE value indicate ...by the fluorescent probe." should be moved to Page 4, the first sentence of Line 172.
  11. Page 9, Line 310-31: "In the presence of ATP and the mutant Tpm actin monomers are deactivated; this can be seen from a decrease in the ΦE value for actin-FITC." This sentence of description is confusing. The authors should explain further about this mechanism as better illustrating their hypothesis, as shown in Figure 4.
  12. Page 11 Line 392-393: The authors indicated that "in the presence of the E139X and A155T mutant Tpms, the myosin heads remain in the conformation of strong binding with actin." However, there is no experiment regarding E139X and A155T mutations in the current study. So this description has no ground.
  13. Page 11, Line 426 & 12, Line 440: The authors mentioned the methods of protein analysis by SDS-PAGE. However, I did not see any results or figures for this experiment. If this is not an essential part of this study, the authors should omit this description.
  14. Page 12 Line 463-467: The description of "an increase in the ΦE value for FITC-actin,...groove of actin (towards the inner domains of actin)." is hard to read and has grammatic errors. The authors should amend it.

Author Response

Reviewer 1

We are very grateful to the Reviewer for the detailed consideration of our work and find the comments fair and constructive. We have addressed each specific point raised by the Reviewer by making additions and clarifications to the text.

Prof. Yurii Borovikov, Dr. Olga Karpicheva

The major criticism for this manuscript:

1.The length of the introduction section should be reduced; otherwise, a large portion of the content should be moved to the Discussion Section.

According to the recommendation of the Reviewer, the Introduction section was reduced. Lines 67-80, 114-123, 131-140, 166-183 on Pages 2-5 were moved to the Discussion section (Lines 408-417, 510-514, 400-403, 423-425, 426-455 on Pages 12-13 of marked version of the manuscript).

2. The “Results and Discussion” section should be separated as Results and Discussion for a better understanding for the readers.

The Results and Discussion section was divided into 2 sections (Pages 4-10, Lines 128-301, and Pages 10-14, Lines 302-490 of unmarked version of the manuscript). We hope this really makes the results easier to understand for the readers.

3. I think in the Introduction Section, the authors should first give a short introduction regarding what did the “ghost fibre” mean? and how did it represent in this study? Should it be more appropriate to be called “modeled“ or “controlled” muscle fibre?

The term and the first method of preparation of “ghost fibres” was introduced in 1976 by Tawada, Yoshida and Morita (J. Biochem., 80, 121-127). The method of preparation used in this study was designed in 1983 by Borovikov and Gusev (Eur. J. Biochem., 136, 363–369). So, we think that it is not appropriate to change the established notion. However, we agree with the Reviewer on the need to give a short introduction what does the “ghost fibre” mean.

Page 3, Lines 105-112:

The aim of the study was to evaluate the effect of the Q147P substitution on the spatial rearrangements of actin, Tpm and the myosin head at some stages of the ATPase cycle. The polarized fluorescence from the probes bound with actin, Tpm or myosin subfragment-1 was registered in ghost fibres – the myosin-Tpm-troponin-free ghosts of skinned single fibres composed of more than 80% actin. Ghost fibres retain the overall highly ordered structure and therefore constitute a perfect model to study the conformational changes of actin and actin-binding incorporated inside exogenous proteins [22]. Thin filaments of ghost fibres were reconstructed with troponin and recombinant mutant or wild-type Tpm.

4.The authors proposed that the negative impact of the mutation may be alleviated with the use of myosin inhibitors. However, I have seen any evidence provide in the current study which supports their postulation. Did the author apply any compound of myosin inhibitor in treating their cells? I suppose that the manipulation of Ca2+ would be more straight-forward to affect this negative impact of the mutation.

Congenital myopathies are rare musculoskeletal diseases; however, they have an extremely negative impact on human life. Until now, there is no treatment aimed at eliminating the primary causes of muscle dysfunction in congenital myopathies. Before testing potential drugs, it is necessary to elucidate in detail the molecular mechanisms of muscle dysfunction with each mutation in tropomyosin, which is what our studies are devoted to. There was a lack of information on the behavior of actin, the myosin heads, and tropomyosin at the molecular level in the presence of the Q147P mutation. Our research has shown that irregular positioning of the Q147P-mutant tropomyosin induces premature activation of actin filaments and a tendency to increase the number of myosin cross-bridges in a state of strong binding with actin at low Ca2+. The next phase of work is an attempt to use some chemical compounds to correct the contractile function of the sarcomere. It is complex work, and we had the first experience of using such compounds in the study of dysfunction caused by tropomyosin carrying the E173A substitution associated with congenital myopathy (Borovikov et al., 2020, Int. J. Mol. Sci. 21, 4421). The skeletal muscle troponin inhibitor W7 (N- (6-minohexyl) 5-chloro-1-napthalenesulfonamide) was used. We found that W7 restores the increase in the number of the myosin heads strongly bound to F-actin at high Ca2+ and stops their strong binding at relaxation, suggesting the possibility of using Ca2+-desensitizers to reduce the damaging effect of the E173A mutation on muscle fiber contractility. It turned out that this approach has been proved to provide very important information that can really be used to treat congenital myopathies.

However, correcting mutation-induced disorders via troponin is much more complicated than it seems. The fact is that currently available troponin inhibitors do not have specificity for binding to skeletal muscle troponin C, but they also bind to the cardiac troponin isoform. Based on existing inhibitors, it is necessary to synthesize a reagent that would be specific to skeletal muscle troponin in order not to disrupt the work of the heart muscle.

In our opinion, it is easier for mutation Q147P to restore the balance between the strongly- and weakly-bound myosin heads during the ATPase cycle using a myosin inhibitor. To mitigate the damage evoked by the Q147P substitution, we assume to use small-molecule inhibitors of myosin, which can specifically suppress the functions of myosin-II – N-benzyl-p-toluenesulfonamide (BTS), butanedione monoxime (BDM) or blebbistatin (Iwamoto, 2018, Biophys. Physicobiol. 15:111-120). BTS specifically binds to skeletal muscle myosin. The information about the therapeutic targets for the restoration of contractile function in the presence of the tropomyosin mutations is needed to initiate the testing of these compounds, primarily in model muscle fibres.

Page 3, Lines 122-127:

The information about the therapeutic targets for the restoration of contractile function in the presence of the tropomyosin mutations is needed to initiate the testing of these compounds, primarily in model muscle fibres. To mitigate the damage evoked by the Q147P substitution, we assume to use small-molecule inhibitors of myosin, which can specifically suppress the functions of myosin-II – N-benzyl-p-toluenesulfonamide (BTS), butanedione monoxime (BDM) or blebbistatin [23].

The minor criticisms for this manuscript:

1. The presentation of Glu147Pro should be "Glu147Pro."

According to the reviewer comment, corrections have been made in the text on Page 3 (Line 97), Page 13 (Line 451), Page 16 (Abbreviations), Page 1 (Lines 3, 16, 24). Also, we corrected a poor spelling (Gln instead Glu).

2. Page 2, Line 90-96: In the era of genetic diagnosis, it has been suggested to do genetic testing (NGS) directly after consulting patients with suspected congenital myopathies, and to put images and invasive muscle biopsy behind the supporting evidence. It seems that the progress of the times can make neuromuscular patients get more invasive tests with less information. Therefore, I think the paragraph should be rewritten to more precisely describe the updated diagnostic algorithm for congenital myopathies, especially the less dependent on the basis of histological changes noted in the biopsy of skeletal muscles. I suggest the authors may refer to this updated article in Lancet Neurol. 2020 19: 522-532.

We are very grateful to the Reviewer for the important reference. The paragraph was rewritten (page 3, lines 90-101) according to the data on the updated diagnostic algorithm for congenital myopathies (Thompson et al., 2020). The appropriate reference was inserted on page 15 (line 540).

Page 2, Lines 77-88:

The correct diagnosis of congenital myopathy variant is critical for both the prediction of the disease course and decision making in the treatment strategy. In clinical practice, up-to-date methods of diagnosis and therapy suggest that genetic testing (NGS) follows the compilation of the clinical history of patient and the initial diagnostic workup [14]. Identification of genetic defect is required primarily due to the heterogeneity of clinical features. However, it has been found that genetic causes are also heterogeneous. A wide range of genetic defects might be responsible for a similar phenotype. Also, the same gene mutation may be attributable to several different skeletal muscle diseases or to unspecified variant of myopathy [15-18]. Thus, NGS approaches should not be used in isolation, but in conjunction with traditional research techniques such as histopathology, biophysics, and biochemistry in an integrated manner [14]. A biopsy of diseased skeletal muscles at congenital myopathy shows different histological changes including the presence of nemaline bodies, rods, cap structures, central cores, minicores, central nuclei and fibre-type disproportion and hypotrophy.    

Page 16, Abbreviations: NGS - next-generation sequencing

          Page 17, References:

  1. Thompson, R.; Spendiff, S.; Roos, A.; Bourque, P.R.; Warman Chardon, J.; Kirschner, J.; Horvath, R.; Lochmüller, H. Advances in the diagnosis of inherited neuromuscular diseases and implications for therapy development, Lancet. Neurol. 2020, 19, 522-532. doi: 10.1016/S1474-4422(20)30028-4.

3. Page 2, Line 83: The description of “even heart failure” is incorrect. This sentence should be rewritten.

The sentence on Page 2 (Lines 67-70) was rewritten:

The consequences of muscle dysfunction vary from mild scoliosis and delayed motor functions to difficulty in moving. Patients with severe disease may suffer from respiratory muscle weakness and respiratory failure. The presence and progression of skeletal deformities can cause severe heart failure [11].

Page 17, References:

  1. Finsterer, J.; Stöllberger, C. Review of Cardiac Disease in Nemaline Myopathy, Pediatr. Neurol. 2015, 53, 473-477. doi: 10.1016/j.pediatrneurol.2015.08.014.

4. Page 2, Line 91: The term "skeletal myopathies" is inappropriate. Should it be congenital myopathies, muscular dystrophies, or metabolic myopathies?

The term "skeletal myopathies" was deleted.

5. Page 3, Line 109 "Congenital myopathy can be regarded as a disease of a thin filament" This description of congenital myopathy is not appropriate, the authors should rewrite it and give a more comprehensive view of the pathomechanism of congenital myopathies.

We agree with the Reviewer that the discussion of the molecular mechanisms of congenital myopathies comprehensively is too complex a question to be briefly introduced in this paper, and we removed the ambiguous sentence. In dependance of the genetic cause, the muscle dysfunction can occur due to the thin filament dysregulation, impaired force-generating capacity of the myosin heads, mitochondrial abnormalities, defects in sarcoplasmic reticulum function, membrane remodeling, neuromuscular junction, etc. (Ravenscroft, Laing, Bönnemann, Brain. 2015; 138(2): 246–268).

It should be emphasized that we are modeling the earliest stages of muscle pathology associated with the mutation in tropomyosin. Our model does not consider the later stages of dysfunction.

6. Page 3, Line 130: Cap myopathy can be caused by mutations in the ACTA1, TPM2, or TPM3 genes. The Q147P mutation belongs to the TPM2 category.

Thank you for the notion. We added the information on Page 3, Line 96-98:

The present study therefore was undertaken to determine the molecular mechanisms of the congenital myopathy caused by one of the mutations in TPM2, leading to Gln147Pro (Q147P) substitution of Tpm2.2. 

7. Page 4 Line 156: Given the fundamental effect on myosin, the authors should more detailely explain the mechanism of "cross-bridges " and its effect. Does it mean "Tpm shifts into the actin groove and ability to move on actin"?

We added the research results at the end of the Introduction section:

Page 3, Lines 116-118:

We found the inability of troponin to switch the thin filaments off at a low Ca2+ concentration and an increase in the number of the myosin cross-bridges strongly bound with actin. The angular orientation of the fluorescence probes bound with Tpm was found to be changed by the substitution and was characteristic for the shift of Tpm strands closer to the inner actin domains. It is proposed that the mutant Tpm can shift deeper into the actin groove and loses the ability to move on actin, which facilitates the binding of myosin cross-bridges to actin.

8. Page 4, Line 166: “decorated” should be corrected as “modified”.

The term “decorated by” was corrected as “modified” (Page 4, Lines 131-133):

Using ghost fibres from rabbit psoas muscle, thin filaments were reconstructed sequentially with exogenous recombinant control (wild-type, WT) Tpm and troponin, and then modified by myosin subfragment-1 (S1).

9. Page 4, Line 172-174: The sentence of "The angles of emission dipoles ..., respectively” is not described correctly and somehow confused to readers, and it should be rewritten for better understanding.

The sentence was rewritten.

Page 4, Lines 133-136:

The polarized fluorescence intensities from actin-FITC, Tpm-AF and S1-AEDANS were registered and the angles of emission dipoles orientation ΦE of the probes, number of disordered fluorophores N (for S1-AEDANS) and bending stiffness (for actin-FITC and Tpm-AF) were calculated.

Page 4, Lines 144-146:

According to the data obtained (Figure 1a,b), in ghost fibres containing Tpm and troponin at a low Ca2+ concentration the values of ΦE for the probes associated with actin (actin-FITC) and Tpm (Tpm-AF) are 46.2 degrees and 61.3 degrees, respectively.

10. Page 5, Line 190-191: The sentence of "The changes in the ΦE value indicate ...by the fluorescent probe." should be moved to Page 4, the first sentence of Line 172.

The sentence was moved (Page 4, Lines 143-144).

11. Page 9, Line 310-31: "In the presence of ATP and the mutant Tpm actin monomers are deactivated; this can be seen from a decrease in the ΦE value for actin-FITC." This sentence of description is confusing. The authors should explain further about this mechanism as better illustrating their hypothesis, as shown in Figure 4.

The section Results (Page 9, Lines 285-297) was supplemented by the additional explanation:

In control, the addition of ATP was accompanied by the decrease in the ΦE value for actin-FITC and by the increase of this value for S1-AEDANS and Tpm-AF as compared with the absence of ATP (Figure 1). Tpm should return in an energetically favorable position over outer actin domains, and actin monomers should be deactivated. It means that actin adopts the conformation that can’t activate ATP hydrolysis in myosin active site. The weak interaction between actin and myosin suggests the decrease in the number of bonds between these two proteins, that was schematically presented in Figure 4. The changes of polarized parameters for the mutant Tpm-AF (decrease in the values of ΦE) in the presence of ATP reveal the aberrant positioning of the mutant Tpm. The decrease in ΦE parameter of polarized fluorescence for actin-FITC was typical for deactivation of actin filaments. Therefore, actin is deactivated in the presence of the mutant Tpm and even more actin monomers transit to the switched-off conformation. Increase in the ΦE value for S1-AEDANS indicates on the transition of the myosin heads to the weak-binding conformational state. Thus, the relaxation process occurs and are not significantly affected by the substitution in Tpm.

12. Page 11 Line 392-393: The authors indicated that "in the presence of the E139X and A155T mutant Tpms, the myosin heads remain in the conformation of strong binding with actin." However, there is no experiment regarding E139X and A155T mutations in the current study. So this description has no ground.

The paragraph was deleted.

13. Page 11, Line 426 & 12, Line 440: The authors mentioned the methods of protein analysis by SDS-PAGE. However, I did not see any results or figures for this experiment. If this is not an essential part of this study, the authors should omit this description.

The description was deleted. The results of the study on the Q147P-mutant Tpm binding with actin (in ghost fibres and in co-sedimentation assay) was presented previously (Karpicheva et al., Biochim. Biophys. Acta 2016, 1864, 260–267).

14. Page 12 Line 463-467: The description of "an increase in the ΦE value for FITC-actin,...groove of actin (towards the inner domains of actin)." is hard to read and has grammatic errors. The authors should amend it.

The sentence was divided into two more simple sentences (Page 16, Lines 558-561):

According to the previous observations [73], an increase in the ΦE value for FITC-actin indicates an increase in the relative amount of switched-on actin monomers. A decrease in the ΦE value for 5-IAF-Tpm and 1,5-IAEDANS-S1 occur when the myosin heads transit to a strong binding with actin and Tpm shiths to the groove of actin (towards the inner domains of actin).

Reviewer 2 Report

The MS deals with the changes in properties of reconstructed ghost muscle fibres caused by a myopathy-associated mutation in the TPM2 gene that causes the Glue147Pro substitution in slow muscle troponin Tpm2.2. Changes in the orientation of fluorescent labels bound to Cys36 and/or Cys190 of Tpm, F-actin, and Cys707 of myosin S1 caused by the addition of Ca2+, troponin S1, ADP, or ATP were estimated using a polarization fluorescence technique. The authors found some changes in the fluorescence polarization responses to the addition of Ca2+ and myosin S1 caused by the G147P Tpm mutation.

I have some concerns and questions that should be addressed before a can estimate acceptance of the MS.

  1. It was shown that the G147P Tmp mutation dramatically reduces the Tmp affinity for actin (Ref. [43]). For this reason, the concentrations of Tmp (WT or G147P) and Tn added to ghost fibers to reconstruct the regulatory function of thin filament should be specified. It looks like some results can be explained by incomplete labeling of thin filament with mutant Tpm. Therefore, control experiments showing that myosin S1 ATPase could indeed be inhibited by Ca2+ removal in reconstructed fibres containing G147P Tpm should be performed.
  2. As the beta-beta-homodimers of Tpm are unstable and beta-Tpm is mainly present in skeletal muscle as the ab-heterodimer (Ref. [43], lines 379-382) the relevance of the result for the understanding and treatment of myopathy caused by the mutation is limited. Do the authors have any data obtained with alpha-beta-Tpm heterodimer?
  3. The Abstract contains statements that do have direct support from data obtained. Lines 19-20: “Compared with wild type, the mutant tropomyosin was found to shift closer to the inner actin domains…” - Fluorescence polarization data provide some information about angular changes with respect to fibre axis, not the radial movement. Lines 21-22: “The Glu147Pro substitution decreases the number of bonds between tropomyosin and actin …” – The authors do not provide any data concerning the number of actin-Tpm bonds. Lines 25-27: “The Ca2+-sensitivity of regulated thin filaments in muscle fibres increases and the functional state of the thin filaments shifts to the open state.” – The authors did not measure the Ca2+-sensitivity of the system, the only compared +/-Ca2+ states. The Abstract should be rewritten to separate what really follows from their data from assumptions.
  4. Lines 202-204: “As was proposed previously [7,49], the changes in flexibility of the filaments associate with the changes in their persistence length.” – “Correlates” is not an adequate term as the persistence length by definition is the bending stiffness, divided by the Boltzmann constant k and absolute temperature T. Please correct.
  5. Lines 198-200: “A decrease in the ΦE value for TM-AF indicates the Tpm shift closer towards the center of the actin filament (to the groove of the actin long helix).” - Changes in any angle measured with fluorescence polarisation are not directly related to Tpm radial movement. Cryo-EM data (Behrmann e.a. Cell, 2012, 150:327-338) suggest that upon myosin S1 binding Tmp moves mainly azimuthally, not radially. Please, be more specific.
  6. Line 212: “The increase in the value of ε correlates with the increase in the persistence length of actin and Tpm filaments.” - See above.

Minor point

Lines 200, 209, 283, 290-291: Change “stiffness bending” for “bending stiffness”.

Author Response

Reviewer 2

We are very grateful to the Reviewer for the detailed consideration of our work and find the comments fair and constructive. We have addressed each specific point raised by the Reviewer by making additions and clarifications to the text.

Prof. Yurii Borovikov, Dr. Olga Karpicheva

1. It was shown that the G147P Tmp mutation dramatically reduces the Tmp affinity for actin (Ref. [43]). For this reason, the concentrations of Tmp (WT or G147P) and Tn added to ghost fibers to reconstruct the regulatory function of thin filament should be specified. It looks like some results can be explained by incomplete labeling of thin filament with mutant Tpm. Therefore, control experiments showing that myosin S1 ATPase could indeed be inhibited by Ca2+ removal in reconstructed fibres containing G147P Tpm should be performed.

The final concentrations of wild-type Tpm, troponin and S1 in modelled fibres was determined in separate experiments previously (Borovikov et al., Biochim. Biophys. Acta, 2009, 1794, 985–994; Borovikov et al., Sci. Rep. 2017, 7, 16797). The experiments conditions did not change. The ghost fibres was incubated in the solution containing 2 mg/ml of the proteins (Tpm, troponin or S1) during 2-3 hours. This duration of the incubation time was enough to achieve the molar ratio of the control tropomyosin and troponin to actin to be 1:6.5 (±2) and 1:6.5 (±2), that was controlled by the SDS-PAGE assays. Also, in the presence of the regulatory proteins the molar ratio of S1 to actin was determined to be 1:5 (±2) in the absence of the nucleotides, 1:4.5 (±2) in the presence of ADP, and 1:14 (±2) in the presence of ATP, respectively. This information was added in Materials and Methods (Page 15, Lines 525-535).

The effect of the change in the amount of the wild-type Tpm in thin filaments of ghost fibre has been investigated when developing experimental protocols. A decrease in the time of incubation of ghost fibre in a solution containing wild-type Tpm induced a decrease in the Tpm concentration in the thin filaments of ghost fibre. By this way, the activation and inhibition effects of the wild-type Tpm at a low and a high Ca2+ on actin-FITC and S1-AEDANS were compared to know how the concentration of Tpm can affect the polarized fluorescence parameters. We did not observe the different effects of Tpm on the polarized fluorescence parameters in dependance on the saturation of thin filaments by Tpm. At a high Ca2+, we registered the changes in parameters of actin-FITC and S1-AEDANS that were typical for activation of actin and for an increase in the pool of S1 strongly bound with actin. At a low Ca2+, the changes in parameters of actin-FITC and S1-AEDANS were typical for switching actin off and for a decrease in the pool of S1 strongly bound with actin. However, the quantitative effect was weaker than in ghost fibres with thin filaments saturated with Tpm.

Indeed, the co-sedimentation assays performed by us (Pronina et al., 2007; Karpicheva et al., 2016) and by our colleagues (Marttila et al., 2012; Matyushenko et al., 2019) showed that the Q147P substitution leads to a weaker binding of tropomyosin with actin. However, the results of co-sedimentation assays do not mean that the mutant tropomyosin cannot incorporate into the thin filaments of skeletal muscle fibers in vivo and work there as a toxic protein. The tropomyosin polymerization on actin is a highly cooperative process. During early stages of thin filament assembly, the weak binding of tropomyosin is expected. At filament maturation and formation of the tropomyosin continuous cable, gaps are eliminated and tropomyosin binds to actin much more strongly. The reviewer is right: if the substitution in tropomyosin weakens the ability to bind well with actin, one can expect the loss of a certain number of tropomyosin molecules and an increase in gaps. However, it turned out that even though the affinity of a mutant tropomyosin for actin reduces dramatically in solution, as was shown for the Gln147Pro (6-7 times) and the Glu139 deletion (5 times) (Marttila et al., 2012)), the content of the mutant Tpm associated with actin in a well-structured system of muscle fibre differs from the control not so significantly (1.7 times for the Gln147Pro and only 1.2 times for the Glu139del mutant Tpms (Karpicheva et al., 2016; Borovikov et al., 2017)). Apparently, during the thin filament assembly the mutant tropomyosin remains bound to thin filaments in the muscle fibre due to the end-to-end associations, wrapping around the F-actin many times and formation of a continuous cable. In this and earlier work tropomyosin of both wild-type and the mutant forms was associated with actin of the ghost muscle fibre by incubation for 2-3 hours in parallel experiments. In the presence of S1, which is known to increase the tropomyosin affinity to actin (Gordon et al., 2000), the values of fluorescence intensities ( Isum= (^I^ + ||I|| + ||I^ + ^I||  ) were similar for both Tpms: 1034 ± 23 for WT Tpm, and 978 ± 11 for the mutant Tpm.

Relative the effect of the mutant Tpm it is worth to note following. It is well known that in the presence of tropomyosin the relative amount of activated actin monomers and the myosin heads strongly associated with actin increases. It seems possible that if the mutant tropomyosin left the muscle fibre area, the amount of activated actin monomers and the myosin heads in strongly-binding conformation would most likely decrease rather than increase. However, in our experiments performed in the absence of troponin (Karpicheva et al., 2016) in the presence of the Q147P-mutant tropomyosin the relative amount of activated actin monomers and the myosin heads in strongly-binding conformation was even greater than in the presence of the wild-type tropomyosin. Thus, it can be considered that the activation of thin filaments in the presence of the mutant tropomyosin is not associated with a decrease in the affinity of the mutant tropomyosin to actin, but is most likely due to the abnormal behavior of tropomyosin on thin filaments and the specific response of actomyosin to this disturbance.

The explanation was added on Page 10 (Lines 332-341), Page 12 (Lines 384-395), Page 16 (Lines 565-577).

2. As the beta-beta-homodimers of Tpm are unstable and beta-Tpm is mainly present in skeletal muscle as the ab-heterodimer (Ref. [43], lines 379-382) the relevance of the result for the understanding and treatment of myopathy caused by the mutation is limited. Do the authors have any data obtained with alpha-beta-Tpm heterodimer?

We have no data obtained with αβ-Tpm heterodimer as this objective was not part of the present work. On the basis of our previous study on the effect of the R133W substitution in the β-chain of ββ-homodimer and αβ-heterodimer (Borovikov et al., Biochem. Biophys. Res. Commun. 2020; 523, 258-262) we suggest, that the investigation of the mutation effect using homodimers is acceptable. We believe that the αβ-heterodimer is a more appropriate model for studying the effects of myopathic mutations in the β-chain of Tpm. The negative effects of the mutation can be partially compensated for by the second tropomyosin chain, which does not contain the mutation. Despite the quantitative differences in the effects, the qualitative nature of those critical changes in the conformational state of actin, myosin, and tropomyosin, which occurred as a result of the amino acid substitution R133W, was the same for the αβ-heterodimer and ββ-homodimer of tropomyosin. The key aspects of impaired contraction regulation were manifested much more clearly with study of homodimers with mutations in both subunits. Thus, the results give an important information about the molecular mechanisms of congenital myopathy associated with the Q147P mutation in TPM2 and, we believe, will help for the testing of potential drugs to eliminate the primary causes of muscle dysfunction. Our point of view was stated on the Pages 13-14, Lines 466-481.

3. The Abstract contains statements that do have direct support from data obtained. Lines 19-20: “Compared with wild type, the mutant tropomyosin was found to shift closer to the inner actin domains…” - Fluorescence polarization data provide some information about angular changes with respect to fibre axis, not the radial movement. Lines 21-22: “The Glu147Pro substitution decreases the number of bonds between tropomyosin and actin …” – The authors do not provide any data concerning the number of actin-Tpm bonds. Lines 25-27: “The Ca2+-sensitivity of regulated thin filaments in muscle fibres increases and the functional state of the thin filaments shifts to the open state.” – The authors did not measure the Ca2+-sensitivity of the system, the only compared +/-Ca2+ states. The Abstract should be rewritten to separate what really follows from their data from assumptions.

The abstract has been amended so that conclusions and assumptions are separated from each other:

Page 1, Lines 19-23:

The angular orientation of the fluorescence probes bound to tropomyosin was found to be changed by the substitution and was characteristic for a shift of tropomyosin strands closer to the inner actin domains. It was observed both in the absence and in the presence of troponin, Ca2+ and myosin heads at all simulated stages of the ATPase cycle. The mutant showed higher flexibility.

Page 1, Lines 23-24:

This data suggest that the mutation decreases the number of bonds between tropomyosin and actin.

Page 1, Lines 25-27:

The sentence was deleted.

4. Lines 202-204: “As was proposed previously [7,49], the changes in flexibility of the filaments associate with the changes in their persistence length.” – “Correlates” is not an adequate term as the persistence length by definition is the bending stiffness, divided by the Boltzmann constant k and absolute temperature T. Please correct.

“Correlate” and “Associate” were changed to “Mean”. Thank you for your notion.

Pages 5-6, Lines 176-179:

As was proposed previously [7,28], the changes in bending stiffness of the filaments mean the changes in their persistence length. Usually, an increase in the persistence length of Tpm strands correlates with a decrease in the persistence length of actin filaments and vice versa.

5. Lines 198-200: “A decrease in the ΦE value for TM-AF indicates the Tpm shift closer towards the center of the actin filament (to the groove of the actin long helix).” - Changes in any angle measured with fluorescence polarisation are not directly related to Tpm radial movement. Cryo-EM data (Behrmann e.a. Cell, 2012, 150:327-338) suggest that upon myosin S1 binding Tmp moves mainly azimuthally, not radially. Please, be more specific.

We agree with the Reviewer that the changes in any angle measured with fluorescence polarization are not directly related to Tpm radial movement. In the present study and some previous works (Borovikov et al., Int. J. Mol. Sci. 2018, 19, 3975; Borovikov et al., Sci. Rep. 2017, 7, 16797), the changes in the ΦE values for 5-IAF-labeled Tpm were considered as being correlated with the azimuthal shifting of the Tpm strands observed at studying the regulation of the actin–myosin interaction by Tpm-troponin and myosin heads in electron microscopy works (Lehman et al., J. Muscle Res. Cell Motil. 2013, 34, 155–163). An increase in the ΦE value correlated with the azimuthal shifting of Tpm strands towards the outer domains of actin subunits, while a decrease in this value correlated with the shifting of Tpm to the inner domains of actin subunits.

The paragraph was changed (Page 5, Lines 162-173):

The FITC-phalloidin binds to the region of three neighboring actin monomers [27], and it can be assumed that an increase in the ΦE value indicates a rotation of actin monomers in the direction from the axis of the thin filament (clockwise). Taking into account current ideas about the regulation mechanisms related to actin, we can conclude that an increase in the ΦE value for actin-FITC is characteristic of actin monomers switching-on. The 5-IAF probe is associated with both cysteine Tpm residues in positions 36 and 190, and the changes in the ΦE value indicate the spatial rearrangements of the Tpm strands on actin. A decrease in the ΦE value for TM-AF was observed at simulation of activated state of thin filaments and is considered as being correlated with the azimuthal movement of Tpm towards the center of the actin filament (to the groove of the actin long helix) showed in electron microscopy works [7]. The Tpm shifting is possible due to an extension or shortening of the actin filament and the Tpm strands along the actin filaments. Based on this conclusion, the effects of the Q147P substitution in Tpm on the spatial rearrangements of actin and Tpm were analyzed below.

6. Line 212: “The increase in the value of ε correlates with the increase in the persistence length of actin and Tpm filaments.” - See above.

The correction was made on Page 6, Line 186:

The increase in the value of ε means the increase in the persistence length of actin and Tpm filaments.

Minor point

Lines 200, 209, 283, 290-291: Change “stiffness bending” for “bending stiffness”.

The corrections were made.

Round 2

Reviewer 2 Report

The authors have responded to my questions and clarified some points in the revised manuscript so that it was significantly improved compared to its first version.

However, some points and statements potentially misleading a reader remain in the revised manuscript. They should be corrected or specified before the manuscript can be accepted.

Also, the major limitations of the work should be specified and summarized in a separate section.

  1. All high-resolution structure of the actin-Tpm complex in the blocked, closed and open states demonstrate only the azimuthal movement of Tpm upon transitions between the three states. Moreover, the method used by the authors can only detect the angular movement of the probes, not their radial shift. To avoid misunderstanding I recommend to changes expression “closer to inner actin domain” with a more specific description of what was observed experimentally in the entire manuscript.
  2. Although the Abstract was rewritten and improved it still contains a statement not following from the data: “This data suggest that the mutation decreases the number of bonds between tropomyosin and actin.” The number was not measured or even estimated.
  3. The authors did not measure the Ca2+ sensitivity of the system. They only tested the “on” and ‘off” states while a shift in the Ca2+-sensitivity is a major component of impairment of Ca2+ regulation caused by a myopathy-associated mutation in Tpm. This limitation should be explicitly explained in the Limitations section.
  4. Discussion concerning the use of beta-beta-homodimers of Tpm instead of alpha-beta-heterodimers which are the major form of Tpm in muscle should be moved to the Limitation section.
  5. The authors suggest using small molecule myosin inhibitors to reduce damage caused by the mutation, although did not teste none of them. This should be stated explicitly.

Minor point

Line 1081: “…we assume to use small-molecule inhibitors…”. ‘Assume’ is not a proper verb, please rephrase.

Author Response

1. All high-resolution structure of the actin-Tpm complex in the blocked, closed and open states demonstrate only the azimuthal movement of Tpm upon transitions between the three states. Moreover, the method used by the authors can only detect the angular movement of the probes, not their radial shift. To avoid misunderstanding I recommend to changes expression “closer to inner actin domain” with a more specific description of what was observed experimentally in the entire manuscript.

As with the majority of studies, the results obtained in the present study were interpreted following some assumptions. We couldn’t summarize the assumptions in the separate section because they were discussed through the text of the manuscript and also in the section Materials and Methods. The additional explanation regarding to the correlation between the changes in the value of ΦE for Tpm-AF and the position of Tpm on actin was inserted in the Discussion section (Page 11, Lines 365-381):

In control, the activation of thin filaments by troponin-high Ca2+ and the myosin heads is accompanied by the increase in the ΦE value for actin-FITC, and by the decrease in this value for Tpm-AF and S1-AEDANS. This pattern of changes has also been observed in our previous studies [3,26,28]. The changes in the value of ΦE for Tpm-AF relative to actin was interpreted according to a widespread theory of the Tpm shifting over actin surface from the outer to the inner actin domains [4]. High-resolution structure of the actin-Tpm complex in the different functional states of thin filament proposes the azimuthal movement of Tpm upon transitions between the states [5]. The polarized fluorescence technique can detect the angular changes of the probe orientation, but not their radial shift [49,50]. However, we found [3,26,29] that the changes in the value of ΦE for Tpm-AF correlate with the movement of Tpm observed by EM studies [5,7,51]. Since the decrease in the value of ΦE for Tpm-AF was observed in the control under activation conditions, then such changes were considered by us as a shift of Tpm strands towards the inner actin domains [3,26,29]. The change in a position of the Tpm strands relative to the inner domain of actin may be associated with a disparity in the alterations in the persistence length of Tpm strands and actin filaments that presumably cause azimuthal shifting of the Tpm strands [6,26,28,29]. For example, if the Tpm strands at transition from the ON to the OFF state undergo a greater elongation than does F-actin, it may cause an azimuthal shift of the Tpm strands towards the outer domain of actin [6,26,28,29,51]. Conversely, a lower compared to F-actin shortening of the Tpm strands move them to the inner domain of actin [6,26,28,29,51].

2. Although the Abstract was rewritten and improved it still contains a statement not following from the data: “This data suggest that the mutation decreases the number of bonds between tropomyosin and actin.” The number was not measured or even estimated.

The misleading sentence was removed.

3. The authors did not measure the Ca2+ sensitivity of the system. They only tested the “on” and ‘off” states while a shift in the Ca2+-sensitivity is a major component of impairment of Ca2+regulation caused by a myopathy-associated mutation in Tpm. This limitation should be explicitly explained in the Limitations section.

On the basis in the character of the changes in the polarized fluorescence parameters of the probes bound to actin, Tpm and S1 under Ca2+ level elevation in the fibre, we proposed that the Q147P mutation decreases the threshold of Ca2+-sensitive activation of the regulatory system. However, the authors did not measure the Ca2+-sensitivity of the system and there is no information about this in the text of the manuscript at all. We correct a sentence concerning the Ca2+-sensitive on Page 14, Line 481-483:

According to the polarized fluorescence data, all three mutations decrease the threshold of the Ca2+-sensitive activation in the spatial rearrangements of actin, the myosin head and Tpm.

4. Discussion concerning the use of beta-beta-homodimers of Tpm instead of alpha-beta-heterodimers which are the major form of Tpm in muscle should be moved to the Limitation section.

The authors are very grateful to the Reviewer for the recommendation to add a discussion concerning the use of beta-beta-homodimers of Tpm instead of alpha-beta-heterodimers to the text of the Manuscript, and it would be illogical to separate this paragraph from the Discussion section.

5. The authors suggest using small molecule myosin inhibitors to reduce damage caused by the mutation, although did not teste none of them. This should be stated explicitly.

The correction was made on Page 1, Lines 27-29 (the sentence was removed) and on Page 14, Lines 506-508:

To mitigate the damage evoked by the Q147P substitution, we assume to use small-molecule inhibitors of myosin that can specifically suppress the functions of myosin-II. The testing of the myosin inhibitor is the objective of the next stage of our research.

Minor point

Line 1081: “…we assume to use small-molecule inhibitors…”. ‘Assume’ is not a proper verb, please rephrase.

‘Assume’ was changed to ‘propose’ on Page 3, Line 126, and on Page14, Line 506.